# The Physiological Functions of AbrB on Sporulation, Biofilm Formation and Carbon Source Utilization in *Clostridium tyrobutyricum*

**DOI:** 10.3390/bioengineering9100575

**Published:** 2022-10-19

**Authors:** Kui Luo, Xiaolong Guo, Huihui Zhang, Hongxin Fu, Jufang Wang

**Affiliations:** 1School of Biology and Biological Engineering, South China University of Technology, Guangzhou 510006, China; 2Guangdong Provincial Key Laboratory of Fermentation and Enzyme Engineering, South China University of Technology, Guangzhou 510006, China

**Keywords:** *Clostridium tyrobutyricum*, AbrB, sporulation, biofilm, carbon catabolite repression

## Abstract

As a pleiotropic regulator, Antibiotic resistant protein B (AbrB) was reported to play important roles in various cellular processes in *Bacilli* and some *Clostridia* strains. In *Clostridium tyrobutyricum*, *abrB* (CTK_C 00640) was identified to encode AbrB by amino acid sequence alignment and functional domain prediction. The results of *abrB* deletion or overexpression in *C. tyrobutyricum* showed that AbrB not only exhibited the reported characteristics such as the negative regulation on sporulation, positive effects on biofilm formation and stress resistance but also exhibited new functions, especially the negative regulation of carbon metabolism. AbrB knockout strain (*Ct/ΔabrB*) could alleviate glucose-mediated carbon catabolite repression (CCR) and enhance the utilization of xylose compared with the parental strain, resulting in a higher butyrate titer (14.79 g/L vs. 7.91 g/L) and xylose utilization rate (0.19 g/L·h vs. 0.02 g/L·h) from the glucose and xylose mixture. This study confirmed the pleiotropic regulatory function of AbrB in *C. tyrobutyricum*, suggesting that *Ct/ΔabrB* was the potential candidate for butyrate production from abundant, renewable lignocellulosic biomass mainly composed of glucose and xylose.

## 1. Introduction

The growth transition from the exponential phase to the stationary phase occurs when microbes face nutrition deprivation or adverse environmental changes, along with the expression of various genes involved in metabolic pathways and adaptive response [1,2]. In sub-optimal environments, some species present physiological characteristics, such as producing extracellular proteases and other degradative enzymes to better utilize surrounding nutrients [3]; secrete antibiotics to compete with nearby microbes [4]; increase biofilm formation to improve stress resistance; undergo a cellular differentiation process to form a dormant spore under the harsh environment [5,6,7]. The expression of such functional genes is controlled by a class of proteins termed global transcription regulator, among which AbrB is one of the well-studied cases.

As a global transcriptional regulator, AbrB was well-reported in *Bacillus* and *Clostridia* strains, which could auto-regulate its expression level during vegetative growth [8] by binding to the targeted promoters in a concentration-dependent manner in *Bacillus subtilis* [9]. The consensus sequence WAWWTTTWCAAAAAAW was firstly identified as the binding sites in *B. subtilis* [10]. Later, a TGGNA motif was identified as the optimal AbrB-binding site, which was recognized in the manner of a general DNA tertiary structure by the N-terminal DNA binding domain of AbrB (residues 1–53) [11,12]. Mutations in Arg 23 and Arg 24 resulted in no detectable DNA binding activity [13]. The C-terminal domain (residues 54–94) was proposed to participate in the assembly of AbrB and the mutation of cysteine residue at position 54 abolished DNA-binding activity in vitro and regulatory activity in vivo [14]. The mutation of Leu 56 and Gln 83 could weaken or destroy the subunit interactions without affecting the DNA binding ability, which indicated that AbrB might consist of a DNA-binding domain and multimerization domain [15]. However, all AbrB homologues in *Clostridium acetobutylicum* lacked the 10-residue DNLKLAGGKL motif that presented in *B. subtilis* [16].

In *Bacillus* species, both AbrB and its ortholog Abh participated in regulating sporulation time [17], gene expressions related to antimicrobials [18], biofilm formation [6] and genetic competence [19]. Moreover, AbrB also participated in the production of bacitracin [20], surfactin [21] and alkaline protease [22]. In *C. acetobutylicum* ATCC 824, there are three *abrB* genes (CAC0310, CAC1941 and CAC3647) playing an important role on the transition between acidogenic and solventogenic phases. The sporulation was delayed but cells and spore morphology were not affected when *abrB310* gene expression was silenced [16]. Disruption of *abrB3647* showed an improved cellular growth and a higher solvent titer, while deletion of *abrB1941* had no effect on growth, acidogenesis and solventogenesis, which suggested that AbrB1941 might not be essential [23]. In *Clostridium perfringens str*. 13, there is only one *abrB* gene in the genome that promotes the formation of the adhered biofilm, which is quite different from the inhibitory effect of *abrB* on the biofilm formation in *Bacillus thuringiensis* and *B. subtilis* [5,24,25]. Therefore, the role of AbrB in Gram-positive bacteria remains ambiguous. The diversity and conservation of AbrB in regulating physiological functions and the production of certain products of strains is of great research significance.

*C. tyrobutyricum* ATCC 25755 is a Gram-positive, strictly anaerobic and non-solvent producing microbial cell factory with a high yield of butyrate [26,27,28]. AbrB in *C. tyrobutyricum* was annotated by whole-genome sequence and proteome analysis [29] and its role on sporulation was predicated by transcriptome analysis [30]. However, it is still unclear whether AbrB affects sporulation, biofilm formation, metabolism or has other functions as reported in *B. subtilis*. 

In this study, the AbrB homologue was identified in *C. tyrobutyricum* by protein homology search and protein functional domains predication. Subsequently, the psychological functions of AbrB in cellular performance, such as sporulation, biofilm and fermentation, were also investigated. This study provides a better understanding of the pleiotropic regulation function of AbrB in *C. tyrobutyricum*, paving the way for a better utilization of *C. tyrobutyricum* for butyrate fermentation.

## 2. Materials and Methods

### 2.1. AbrB Sequence Alignment and Functional Domain Prediction in C. tyrobutyricum ATCC 25755

The genome information of the reference strain *B. subtilis str*. 168, retrieved from the NCBI website (GenBank accession no. GCA_000009045.1), was used for sequence analysis of the AbrB homology searches. Amino acid sequences were retrieved from the NCBI database and the protein homology search was performed with BLASTP, available at NCBI (http://blast.ncbi.nlm.nih.gov. accessed on 12 March 2022). Multiple amino acid sequences were aligned with Clustal W using default parameters [31], and the phylogenetic tree was constructed with the Neighbor-Joining method in software MEGA-X [32]. The SMART database (http://smart.embl.de. accessed on 12 March 2022). was employed for predicting protein functional domains [33].

### 2.2. Strains and Culture Conditions

*Escherichia coli* CA434 were used for plasmids propagation and as the donor strain for conjugation [34]. *C. tyrobutyricum* ATCC 25755 was used as the host strain [27]. Strains, plasmids and primers used in this study are listed in Appendix A.

*E. coli* CA434 was aerobically cultured in Luria-Bertani medium (LB) or LB agar plates at 37 °C supplemented with 25 μg/mL chloramphenicol (Cm) and/or 50 μg/mL kanamycin (Kan) when needed. *C. tyrobutyricum* ATCC 25755 and its mutants were anaerobically cultivated at 37 °C in serum bottles containing Reinforced *Clostridial* Medium (RCM) or *Clostridium* Growth Medium (CGM) [35]. The concentration of 25 µg/mL thiamphenicol (Tm) and/or 40 μg/mL Erythromycin (Em) were supplemented for selecting *C. tyrobutyricum* mutants. Antibiotics were purchased from Sangon Biotech, of which the purity of chloramphenicol, kanamycin and thiamphenicol was USP Grade, and erythromycin was Pharmaceutical Grade. All medium was sterilized at 115 °C for 30 min.

### 2.3. Plasmids Construction and Transformation

The deletion of the *abrB* gene was performed according to previous study [36]. P*lac*, the lactose inducible promoter, and the repeat sequence were amplified using primers P*lac*-F/P*lac*-repeat-s-R from the pcloneEZ-Plac-repeat plasmid. The repeat-terminator fragment was amplified from the pcloneEZ-repeat-terminator plasmid by the primers spacer-pb-F/spacer-term-R. The P*lac*-repeat-spacer-repeat-terminator fragment was obtained by overlapping PCR of the P*lac*-repeat and repeat-terminator using the primers P*lac*-F/spacer-term-R. Two homology arms, H1 and H2 (~500 bp each) of the *abrB* gene, were obtained by PCR amplification from genomic DNA of *C. tyrobutyricum* by primers H1-F/H1-R and H2-F/H2-R. Primers H1-F/H2-R were used to overlap H1 and H2 fragments and H1-H2 was obtained. The two fragments described above (P*lac*-repeat-spacer-repeat-terminator and H1-H2) were finally assembled and inserted into *Not*I/*Aat*II sites of pMTL83151 (BBI, Shanghai, China), named pM3/*ΔabrB* (Appendix A).

pMTL82151-Em was used for overexpressing the *abrB* gene. pMTL82151-Em was firstly obtained by replacing the chloramphenicol resistance on plasmid pMTL82151 with erythromycin resistance. P*cat*1, one constitutive promoter, and the *abrB* gene were obtained by PCR amplification from genomic DNA of *C. tyrobutyricum* with primers P*cat*1-F/P*cat*1(*abrB*)-R and *abrB*-F/*abrB*-R, respectively. Then, the plasmid pMTL82151-Em/P*cat*1-*abrB* was constructed by inserting the *cat*1 promoter and *abrB* gene into *Not*I/*Aat*II sites of pMTL82151-Em (Appendix A).

Recombinant plasmids were transformed into *E. coil* CA434 for amplification and confirmed by sequencing (Tianyi Huiyuan, Guangzhou, China), then extracted with the FastPure^®^ Plasmid Mini Kit (Vazyme, Nanjing, China). Recombinant plasmids were transformed into *C. tyrobutyricum* ATCC 25755 by conjugation [35]. The *Ct/ΔabrB* strain was then obtained after being induced in an RCM plate containing 25 mM lactose. The *abrB* complement strain was obtained by introducing the pMTL82151-Em/P*cat*1-*abrB* plasmid into the *Ct/**ΔabrB* strain, named *Ct/ΔabrB-abrB*. Engineering strains were confirmed by colony-PCR with Hieff Canace^®^ Gold Master Mix (Yeasen, Shanghai, China). Added to 1.5% agarose gel was 5 μL of PCR product that was run for 15 min at 120 V by gel electrophoresis (Bio-Rad, Hercules, CA, USA). 

### 2.4. Biofilm Formation Assay

Biofilm formation was measured as previously described with some modifications [37]. Pre-cultures of *C. tyrobutyricum* ATCC 25755 and mutants were transferred into fresh RCM medium until the early stationary phase (OD_600_ = 3.5–4.0). Then, 20 μL cultures were diluted to 50-fold on 48-well polystyrene plates and cultured for 2 days. Next, supernatant and non-adherent cells were carefully removed and washed once with phosphate-buffered saline (PBS). Subsequently, 1.5% (*w*/*v*) crystal violet (CV) was added to stain cells for 30 min and washed twice. Finally, the bound crystal violet was solubilized with 95% ethanol and the absorbance at 595 nm was measured. All experiments were performed in triplicate.

### 2.5. Sporulation Assay

*C. tyrobutyricum* ATCC 25755 were induced to sporulate after 3 days cultivation in RCM and heat-resistant spore formation was measured as previously described [7]. Strains were collected and thermally stimulated at 80 °C for 10 min and then diluted to plate on RCM plates for 3 days cultivation, anaerobically.

### 2.6. Stress Tolerance Assessment and Plate Survival Assay

*C. tyrobutyricum* ATCC 25755 and the mutant were anaerobically cultured in RCM medium at 37 °C and 150 rpm with 2.5% (*w*/*v*) NaCl, or at pH 4.5 with HCl addition or with 10 g/L butyrate to create the hyperosmotic stress, acid shock or product culture medium, respectively. The serum bottles were sparged with nitrogen to achieve anaerobic conditions with (5%, *v*/*v*) inoculation to evaluate product resistance, osmotic pressure tolerance and acid shock tolerance. Samples were taken at regular time intervals to monitor the growth conditions. OD_max_, the optical density at 600 nm with the largest value, was used to evaluate the growth of the strain. μ_max_, the specific growth rate, was obtained by calculating the optical density increase rate during the exponential phase. Each experiment was performed in duplicate.

For plate survival assay, agar was firstly added in RCM medium with different culture conditions, as described above, and sterilized. Secondly, *C. tyrobutyricum* ATCC 25755 and mutants were incubated to the late log phase. Then, 3 OD_600_/mL cells were diluted to 10^0^, 10^−1^, 10^−2^, 10^−3^ and 10^−4^ in sequence. Finally, 5 µL diluted cells were pipetted on the pre-deoxygenated RCM plates according to a previous study [38] and cultured at 37 °C for 2 days in the anaerobic chamber (SHELLAB, Cornelius, OR, USA). 

### 2.7. Quantitative Real-Time PCR (qRT-PCR)

Primers used for qRT-PCR are listed in Appendix A. *C. tyrobutyricum* and mutants were cultured in RCM medium at 37 °C and 150 rpm at the specific time. Total RNA was extracted using a FastPure^®^ Cell/Tissue Total RNA Isolation Kit V2 (Vazyme Biotechnology, Nanjing, China). Then, 500 ng RNA was used to synthesize cDNA by reverse transcription with an ABScript™ RT Reagent Kit (Vazyme Biotechnology, Nanjing, China). TB Green^®^ Premix Ex TaqTM II (TaKaRa, Dalian, China) was used for qRT-PCR (LightCycler^®^ 96, Roche, Shanghai). PCR reactions consisted of an enzyme activation step for 30 s at 95 °C, 40 cycles of denaturation at 95 °C for 5 s, annealing and extension at 60 °C for 30 s, followed by a final extension at 72 °C for 30 s and melt curve analysis. After qRT-PCR, the relative quantitation of mRNA expression was normalized to the level of constitutive expression of the *thl* gene and data analysis was carried out with LightCycler96 SW1.1 software.

### 2.8. Fermentation Kinetics Studies

Substrate utilization profiles of *C. tyrobutyricum* ATCC 25755 and mutants were firstly evaluated with glucose, xylose, fructose, mannose and mannitol in CGM buffered with 40 g/L CaCO3 at 37℃ and 150 rpm in serum bottles, respectively. Batch fermentation kinetics was conducted with 40 g/L glucose/xylose (1:1), 40 g/L glucose/mannose (1:1) and 40 g/L glucose/mannitol (1:1) in CGM buffered with CaCO3. Samples were taken at regular time intervals and kept at −20 °C before HPLC analysis. Each experiment was performed at least twice.

### 2.9. Analytical Methods

Cell density was monitored by measuring the optical density (OD) at 600 nm with a spectrophotometer (PERSEE T6, Beijing, China). Sugars consumption (glucose, xylose, fructose, mannose and mannitol) and organic acids production (butyrate, acetate and lactate) were determined by HPLC (Waters, Milford, CT, USA) with an Aminex HPX-87H Column (Bio-Rad, Hercules, CA, USA) at 60 °C and a refractive index detector (Waters, Milford, CT, USA) at 40 °C. In addition, 2.5 mM H_2_SO_4_ solution at 0.6 mL/min was used for the mobile phase.

## 3. Results

### 3.1. Multiple Sequence Alignment and Functional Domain Prediction of AbrB

The initial identification of AbrB in *C. tyrobutyricum* was based on amino acid sequence homology with some *Bacilli* and *Clostridia* strains. As shown in Figure 1A, the total similarity of AbrBs in amino acid sequence was 67.50%, indicating high conservation among spore-forming Gram-positive strains. The first 54 residues in the N-terminal domain, which were responsible for DNA binding, exhibited high similarity, especially with Arg23 and Arg24 (green pentagram) conserved in all sequences. The SpoVT-AbrB domain was identified by SMART, located from the eighth amino acid to the fifty-first amino acid in *C. tyrobutyricum* ATCC 25755. This domain was found in AbrB from *B. subtilis* [39]. In addition, the N-terminal domain of AbrB in *C. tyrobutyricum* was predicted to contain the αββ structure, which was also found in *B. subtilis* [13]. In contrast, the C-terminal amino acids functioning in AbrB homo-tetramerization in *B. subtilis* exhibited a lesser degree of similarity compared to that of in *Clostridia*. For example, a 10-residue DNLKLAGGKL motif presented in *B. subtilis* was missing in *Clostridia*; however, the cysteine residue at position 54 was still conserved among all AbrBs. Therefore, these results indicated that diversity of C-terminal amino acids might be the main reason for the unique function of AbrB in different strains.

To evaluate the evolutionary relationship of AbrB in spore-forming Gram-positive strains, previous reported amino acid sequences of AbrB were aligned and used for phylogenetic analysis (Figure 1B). Based on this, AbrBs sequences were assigned into three clusters. Cluster I contained AbrBs of *Clostridia* with high sequence similarity to AbrB from *C. tyrobutyricum*, exerting an important influence on acidogenesis and solventogenesis in *C. acetobutylicum* [23] and adhered biofilm formation in *C. perfringens* [24]. AbrBs of *Bacilli* were divided into two distinct clusters. AbrBs in *Bacillus cereus* and *B. thuringiensis* formed cluster II along with that of *B. subtilis* 168. Moreover, BCAH187_A2217 of *B. cereus* formed the unique cluster III with HD73_0390 of *B. thuringiensis*, sharing the least amino acid similarity with AbrB from *C. tyrobutyricum*.

### 3.2. Phenotypic Analysis of the AbrB Mutants

To address the function of AbrB in *C. tyrobutyricum*, the *abrB* gene was deleted or overexpressed, respectively. The *Ct/ΔabrB* strain (Appendix A lane 2) was generated by homologous recombination (Appendix A) resulting in a gene gap of 246 bp in the genome compared with *Ct* (Appendix A lane 1). The presence of the P*cat*1-*abrB* fragment was confirmed by PCR in the *Ct/abrB* strain (Appendix A lane 3).

AbrB could regulate gene expression in the exponential phase; thus, the expression level of the growth phase-specific *abrB* gene was determined by qRT-PCR in *C.tyrobutyricum*. At the end of the lag phase (4 h), the expression level of the *abrB* gene was relatively low. Until the middle (8 h) and the end (10 h) of the exponential phase, an increase of 4.6-fold and 7.9-fold was observed for AbrB expression, respectively (Figure 2A). Finally, the transcription level of the *abrB* gene sharply decreased within 14 h. The high expression level of the *abrB* gene in the exponential phase was necessary for the function of AbrB. Previous studies showed that AbrB homologs in *C. acetobutylicum* had a significant effect on strain growth and the effect of AbrB on growth analysis was conducted. However, there was no significant growth differences in *Ct* and two mutants (Figure 2B).

### 3.3. Negative Regulation of AbrB on Sporulation

Generally, microbes could resist adverse environments and then improve survival by forming highly stress-resistant spores under extreme conditions. The formation of spores needed Spo0A, the master regulator of sporulation, and sigma factors sequential activation. Spo0A was initially transcribed from a σ^H^-dependent promoter and then phosphorylated by orphan histidine kinases [40].

In *B. subtilis*, AbrB could negatively regulate the expression of σ^H^ [41] and *spo0A* genes [42]. As seen in Figure 3A, the spores of the *Ct/∆abrB* strain was approximately 3 × 10^5^ CFU/mL, with three-time increase compared to that of *Ct*. In addition, the spores of the *Ct/abrB* strain decreased by 20% compared with *Ct*. These results indicated that the *abrB* gene had an obvious negative regulation on sporulation.

To explore the molecular mechanism of AbrB on spore formation, the expression levels of *spo0A* and five sigma factors were measured in *Ct/ΔabrB* and *Ct/abrB* strains. Not surprisingly, these genes were all significantly up-regulated in the *Ct/ΔabrB* strain and down-regulated in the *Ct/abrB* strain, respectively. The expression levels of the *spo0A* and *sigH* gene, especially, were increased by 3.5 and 11.0 times in the *Ct/ΔabrB* strain compared with *Ct*, respectively (Figure 3B). The expression level of four sporulation-specific sigma factor genes (*sigF*, *sigE*, *sigG* and *sigK*), which were controlled by Spo0A-P and σ^H^ also significantly improved more or less. The sequential activation of σ^F^, σ^E^, σ^G^ and σ^K^ was important in sporulation, but the expression order was strain-dependent [40]. The involvement of SigK in early-stage sporulation was found in *C. acetobutylicum* and *C. botulinum* [43,44]. To explore whether SigK also had an early role during sporulation in *C. tyrobutyricum*, four sigma factors were deleted (Appendix A) and the expression level of *spo0A* and sigma factors was measured. Except for *sigK*, knocking out other sigma factor genes down-regulated other sigma factors expression levels (Appendix A) and up-regulated *spo0A* and *sigH* expression, while the expression levels of *spo0A* and other upstream sigma factors were all significantly decreased in the *Ct/ΔsigK* strain. The result suggested that SigK might also have an early role in the sporulation progress of *C. tyrobutyricum*.

The spores formation not only needs the *spo0A* gene and sigma factors, but also requires other genes involved in physiological functions, such as coat proteins [45], regulatory factors [46] and cell membrane synthesis [47]. Lipids and proteins are the two main components of the cell membrane. Based on whole-genome sequence [29] and KEGG [48], the gene locus related to the lipid synthesis pathway were analyzed and are shown in Figure 3C. In addition, the transcriptional level of genes related to the fatty acid synthesis pathway was measured, which catalyzed malonyl-CoA to fatty acyl-ACPs (FA), as mentioned in a previous study [47]. The expression of *fabF*, *fabG* and *fabK* were significantly decreased by 41%, 35% and 37% in the *Ct/abrB* strain compared with *Ct*, while the inactivation of *abrB* resulted in 2.5-, 2.1-, 2.5- and 2.1-fold increases in *fabD*, *fabF*, *fabG* and *fabZ* expression, respectively (Figure 3D). The high expression level of *fab* genes implied that the FA synthesis could also be improved in the *Ct/abrB* strain. Additionally, FA is the raw material for the synthesis of phospholipids, an important component of membrane biogenesis [47]. Enhanced lipid biosynthesis suggested that the formation of membranes might also be improved in the *Ct/ΔabrB* strain. The formation of membranes was important in the stages of asymmetric division and forespore engulfment during sporulation development.

### 3.4. Positive Regulation of AbrB on Biofilm Formation

Biofilm is a multicellular community that consists of microbial cells and an extracellular polymeric substance matrix [49], which improves the viability of bacteria. In *B. subtilis*, *abrB* and *spo0A* genes were confirmed to regulate biofilm formation [50]. Numerous experiments have been conducted to explore the influence of AbrB on biofilm formation in *Bacillus* strains, but little attention has been paid to the influence in *Clostridia* strains. To determine the molecular mechanism of biofilm formation in *C. tyrobutyricum*, the biofilm biomass of *abrB* and *spo0A* mutants were quantified.

As shown in Figure 4A, knockout of the *abrB* gene resulted in an impaired biofilm, while the biofilm formation was strengthened in the *Ct/abrB* strain. However, the deletion of the *spo0A* gene seemed to have little effect on the crystal violet staining result. When both *abrB* and *spo0A* genes were knocked out, the biofilm formation level of the strain was significantly decreased compared with the *Ct/∆abrB* strain. CV associated with the adhered cells in 48-well polystyrene plates was solubilized and quantified by measuring absorbance at 595 nm (Figure 4B). It showed that AbrB positively regulated biofilm formation in *C. tyrobutyricum*. However, knockout of *spo0A* improved the biofilm formation without significance (Figure 3B showed that *abrB* overexpression decreased *spo0A* transcription levels). This implied that AbrB could directly regulate biofilm formation rather than Spo0A, which was different from that of *B. subtilis*. The effect of sigma factor deletion on biofilm formation was measured. Figure 4D shows that the deletion of *sigF*, *sigE* and *sigK* had no obvious effect on biofilm formation, while the *Ct/∆sigG* strain showed increased biofilm levels.

In summary, AbrB exhibited a positive regulation on biofilm formation in *C. tyrobutyricum*. At the same time, the biofilm formation may also be regulated by the SigG factor (Figure 4C,D), which was regulated by Spo0A. The contradictory regulation effect of AbrB on biofilm formation in different strains may be owing to the difference in amino acid sequence, which needs further investigation.

### 3.5. AbrB could Enhance the Stress Resistance of C. tyrobutyricum

As mentioned above, AbrB could positively regulate biofilm formation, so it seems that resistance ability against different environments could be strengthened, such as high product concentration, hyperosmotic stress and acid shock [49]. To explore the effect of AbrB on stress resistance, the relation between biofilm formation and stresses resistance were determined.

Results indicated that deleting the *abrB* gene had no significant effect on growth, but the specific growth rate was significantly lower in the *Ct/∆abrB* strain than that of *Ct* and *Ct/abrB* strains (Figure 5A). When cultured under different adverse environments, the OD_max_ and μ_max_ of *Ct/∆abrB* were always significantly lower compared with *Ct* or *Ct/abrB*. It should be noted that the overexpression of *abrB* did not significantly improve the growth when exposed to 2.5% NaCl or pH 4.5 adverse environments. Under the culture condition of 10 g/L butyrate, the overexpression of *abrB* significantly increased the OD_max_ and μ_max_ compared with *Ct*. It could be concluded that *abrB* deletion damaged the resistance to different adverse environments, while the overexpression of *abrB* could improve tolerance to different environmental stresses. Subsequently, plate assays showed that the *Ct/abrB* strain could grow better than *Ct* under various conditions. However, the viability of the *Ct/∆abrB* strain was significantly inhibited, especially in the hyperosmotic and high product culture (Figure 5B).

In the meantime, the differences in stress tolerance could also be observed in biofilm formation ability (Figure 5C). Results showed that the biofilm formation of the *Ct/∆abrB* strain was significantly impaired in all adverse environments mentioned above and the *Ct/abrB* strain had the highest biofilm formation level. To sum up, AbrB was very important in enhancing cell resistance to the adverse circumstances tested above. This stress resistance was closely related to the ability of biofilm formation.

### 3.6. Effect of AbrB on the Fermentation Performance of C. tyrobutyricum

Disruption of AbrB homologs could have a significant effect on the biphasic fermentation of *C. acetobutylicum*. For instance, the deletion of *abrB0310* greatly impaired glucose consumption and solvent production, while the deletion of *abrB3647* enhanced glucose consumption and ethanol production, but solventogenesis was hardly affected [23]. However, it is unclear whether AbrB would also affect carbon source utilization in *C. tyrobutyricum*.

To evaluate the effect of the *abrB* gene on substrates utilization, glucose, xylose, fructose, mannose and mannitol were examined (Figure 6). When glucose was used as the sole carbon resource, no significant difference in sugar consumption rate was observed between *Ct* and *Ct/∆abrB* strains (Figure 6A,B). However, butyrate production and productivity were significantly decreased in the *Ct/∆abrB* strain compared with *Ct* (Table 1). Knockout of *abrB* slowed down the rate of fructose utilization and butyrate production (Figure 6C,D). The fructose consumption rate and butyrate productivity were decreased by 32% (0.77 g/L·h vs. 0.58 g/L·h) and 48% (0.28 g/L·h vs. 0.19 g/L·h), respectively (Table 1). However, there was no significant difference in the final concentration of fructose and butyrate. When mannose or xylose was used as substrate, respectively, *abrB* knockout impaired the carbon source utilization of *C. tyrobutyricum* with decreased butyrate production. When ~52 g/L mannose was used as the sole substrate, *Ct* produced 5.76 g/L butyrate from 11.85 g/L mannose within 72 h (Figure 6E), while *Ct/∆abrB* could only utilize approximately half the mannose (6.60 g/L) and produce half the butyrate (3.05 g/L) (Figure 6F). For *Ct/∆abrB* (Figure 6H), it produced 13.71 g/L butyrate from 32.16 g/L xylose, a decrease of 22% to the control strain *Ct* (Figure 6G). Different from the effects on carbon source utilization mentioned above, knockout of *abrB* enhanced mannitol utilization. *Ct* could only uptake 5.09 g/L mannitol and produced 1.05 g/L butyrate with 1.26 g/L lactate (Figure 6I). However, for *Ct/∆abrB*, significant increase (226%) in butyrate concentration was observed since more mannitol (113%) was consumed and no lactate was produced (Figure 6J, Table 1).

Glucose, xylose, mannose and mannitol were major fermentable sugars in most of the lignocellulose hydrolysates [27], softwood hydrolysates [51] and macroalgal hydrolysates [52]. To investigate the effect of AbrB on mixed carbon source utilization, glucose/xylose, glucose/mannose and glucose/mannitol were examined. The deletion of *abrB* showed no significant difference on sugar consumption when the glucose/mannose mixture was used as the carbon source (Figure 7A,B); however, acetate concentration peaked at 24 h, followed by re-assimilation with a slight increase (13%) for butyrate production in the *Ct/∆abrB* strain (Figure 7B). *Ct* was able to co-utilize glucose and mannitol, achieve glucose depletion within 24 h and mannitol within 36 h. At the end of fermentation, *Ct* converted 16.25 g/L glucose and 20.15 g/L mannitol to 14.95 g/L butyrate with trace by-product lactate and acetate (Figure 7C). Mannitol utilization decreased significantly (75%) and the by-product lactate production increased significantly (172%) in *Ct/∆abrB* compared with *Ct* (Figure 7D). 

Especially when the glucose/xylose mixture was used as substrates, glucose was quickly depleted by the *Ct* strain within 24 h, with only trace consumption of xylose (7.44%), leading to the final titer of 7.01 g/L butyrate and low productivity of 0.02 g/(L∙h). It showed obvious Carbon Catabolite Repression (CCR) between glucose and xylose in *C. tyrobutyricum* (Figure 7E), which was consistent with a previous report [27]. However, *abrB* deletion exhibited satisfactory fermentation performance. Interestingly, 59.97% of xylose consumption was absorbed by the *Ct/ΔabrB* strain (Figure 7F), which produced 13.11 g/L butyrate with the productivity of 0.19 g/(L∙h) (Table 1). 

## 4. Discussion

*C. tyrobutyricum* is a promising butyrate producer with a well-illustrated metabolic pathway and convenient genetic tools [36,53,54], which could serve as an excellent platform for valuable biochemicals production from renewable and cheaper feedstocks [55,56,57,58,59]. Whole-Genome sequence and proteome analysis showed that there was one putative *abrB* gene in the genome of *C. tyrobutyricum* [29]. After overexpression of a heterologous uptake hydrogenase in *C. tyrobutyricum*, the role of AbrB in regulation on sporulation process was hypothesized by comparative transcriptome analysis [30]. However, the function of the pleiotropic regulator AbrB in *C. tyrobutyricum* has not yet been studied in-depth.

The function of AbrB was confirmed by sequence alignment and functional domain prediction based on the result of genome sequencing [29]. Results of multiple sequence alignment and phylogenetic analysis showed that AbrB in *C. tyrobutyricum* had 51.04% similarity in amino acid sequence compared to *B. subtilis* 168 (BSU00370). Although there were obvious differences in the amino acid sequences of AbrBs between *Bacillus* and *Clostridia* strains, the functional domain and amino acid residues were highly conserved (Figure 1A). The clustering of AbrB protein on the evolutionary tree might imply that the function of AbrB could be different between different genera while conservative in the same genus. It should be noted that the amino acid sequence similarity of AbrBs in the same genus might be related to its function. AbrB 1941, with the lowest sequence similarity to AbrB from *C. tyrobutyricum* among all *Clostridia* AbrBs (Figure 1B), was verified to not necessarily being essential in *C. acetobutylicum* [23].

The psychological function of AbrB was explored and summarized in Figure 8. Generally, AbrB regulated the sporulation process by regulating the transcription of *spo0A* and sigma factors and affecting the composition of the cell membrane during the stages of asymmetric division and forespore engulfment. In *C. tyrobutyricum*, *abrB* was highly transcribed at the end of the exponential period and it showed no obvious effect on cell growth with knockout or overexpression (Figure 2). However, *abrB* deletion could significantly increase the number of spores (Figure 3A). The regulation mechanism of AbrB on sporulation was further verified by qRT-PCR (Figure 3B,D), in which the expression level of *spo0A*, sigma factors and the *fab* operon were increased with *abrB* knockout. The negative regulation of AbrB on sporulation was consistent with that of *Bacilli* strains [42]. However, the deletion of *abrB* impaired the biofilm formation (Figure 4A,B), which was opposite to that of *Bacilli* strains. It was reported that AbrB could inhibit biofilm formation in B. *subtilis* by regulating a putative secreted protein YoaW and a signal peptidase SipW [5]. The biofilm formation of *B. amyloliquefaciens* and *B. thuringiensis* was enhanced when the *abrB* gene was disrupted [6,25]. With overexpression of the *abrB* gene, the adhered biofilm formation was improved in *C. perfringens* [24], which was consistent with our study in that AbrB showed positive regulation on biofilm formation. It is reported that the disruption of *spo0A* would increase the expression of *abrB* and deprive the ability of biofilm formation in *C. acetobutylicum* [60], which was contrary to our study in that AbrB positively regulates the biofilm formation in *C. tyrobutyricum*. It revealed that the effect of *abrB* on biofilm formation might be strain-dependent [1]. The biofilm formation of *Ct/**Δspo0AΔ**abrB* was significantly lower than that of *Ct/**Δ**abrB*, suggesting that the biofilm of the *C. tyrobutyricum* is not only directly regulated by AbrB, but also indirectly controlled by an unknown gene regulated by Spo0A. The σ^E^ pathway was found to be activated during biofilm formation and the inhibition of *rpoE* expression led to a significant decrease in the biofilm ability of *E*. *coli* [61]. The biofilm formation level of *Ct/**Δ**sigG* was significantly improved compared to *Ct,* while the other three *Ct/**Δ*sigma mutants had no significant effect on biofilm formation. These results showed that AbrB could regulate the biofilm formation directly and enhance the biofilm formation indirectly by repressing the expression of the SigG factor. The high level of biofilm improved the tolerance of *Ct/abrB* to different adverse environments (Figure 5). However, whether there are other genes that affect biofilm formation is a focus that needs to be further studied.

The regulation of *abrB* on glucose consumption and solvent production was found in *C. acetobutyricum*. When *abrB310*, a highly similar homolog to *abrB*, was silenced by antisense RNA, the acids production was accumulated to approximately two-fold higher than wild-type with the solventogenesis formation delay [16]. Deletion of another *abrB* homolog, *abrB3647,* resulted in better sugar consumption and faster solvent production in *C. acetobutyricum* [23]. *abrB* deletion showed a diversity of effects on the utilization of different kinds of carbon sources in *C. tyrobutyricum* (Figure 6). Among tested single carbon sources, *abrB* deletion does not affect the utilization of glucose or fructose (Figure 6A–D). However, mannose or xylose assimilation was greatly damaged while mannitol absorption was enhanced in *Ct/**Δ**abrB* (Figure 6E–J). The fermentation performance of *Ct/**ΔabrB* with glucose/mannose and glucose/mannitol was not very satisfactory. When cultured under 40 g/L glucose/xylose (1:1), *abrB* deletion was found to improve xylose utilization and could alleviate the CCR between glucose and xylose in *C. tyrobutyricum* (Figure 7E,F); thus, it promoted butyrate production compared with *Ct/ΔabrB* (Table 1). Glucose and xylose are the two major fermentable sugars in most of the lignocellulose hydrolysates [62]. However, the utilization of xylose was severely impaired due to the carbon catabolite repression (CCR) mediated by glucose, which was the main obstacle for the industrial production of butyrate [27]. Based on the above results, the deletion of *abrB* could alleviate the CCR between glucose and xylose in *C. tyrobutyricum* to some extent, indicating that the *Ct/ΔabrB* strain seemed to be the potential candidate for butyrate production from cheaper lignocellulosic feedstocks.

It should be noted that the effect of AbrB on the utilization of xylose was opposite when xylose and glucose/xylose was used as the carbon source. *abrB* deletion impaired the utilization of xylose when xylose was used as the sole carbon resource (Figure 6G,H). However, *Ct/ΔabrB* could enhance the utilization of xylose after glucose depletion under a glucose/xylose mixture as a carbon resource (Figure 7E,F). The opposite effect of AbrB on carbon source utilization under different types of carbon sources could also be observed in mannitol. The mechanism behind this phenomenon needs to be further explored.

## 5. Conclusions

In conclusion, this study revealed functions of AbrB on sporulation, biofilm formation, stress resistance and fermentation in *C. tyrobutyricum*, endowing the *Ct/abrB* strain with the stronger ability to resist the adverse conditions, such as high substrate concentration, hyperosmotic stress and acid shock. Additionally, the deletion of the *abrB* gene could alleviate the CCR between glucose and xylose in *C. tyrobutyricum* to some extent. This study extended the understanding of AbrB in *C. tyrobutyricum* ATCC 25755.

## Figures and Tables

**Figure 1 bioengineering-09-00575-f001:**
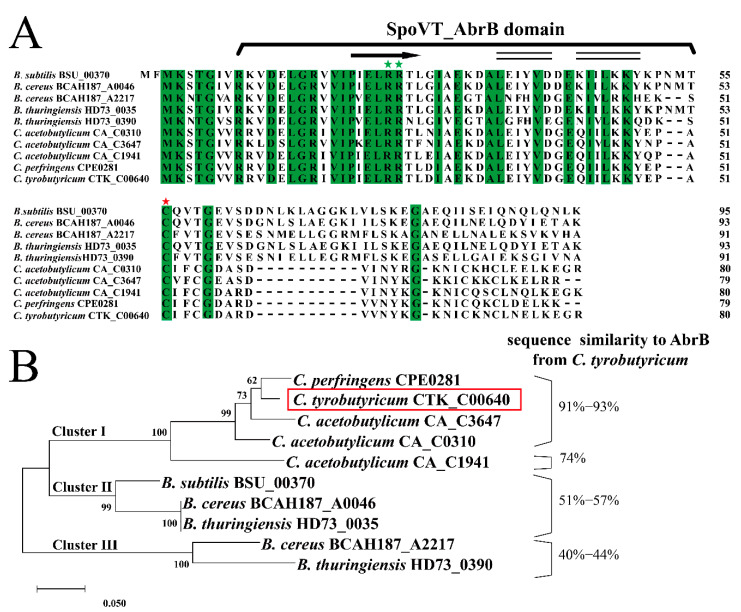
Identification and analysis of AbrB in *C. tyrobutyricum* ATCC 25755. (**A**) Amino acid sequence alignment of AbrB (CTK_RS00405) in *C. tyrobutyricum* ATCC 25755 with the AbrB homologs from other selected strains. The residues predicted to encode the SpoVT-AbrB domain are identified, and the identical residues are highlighted in green. The conserved cysteine acid residue responsible for AbrB polymerization and DNA binding activity is marked with red stars and the conserved arginine residues for DNA binding are marked with green stars. Secondary structure of AbrB was predicted according to NMRPROCHECK and NOEs (α-helix, black double-line; β-sheet, black arrows). (**B**) Maximum-likelihood protein similarity tree based on amino acid sequences of AbrB from *C. tyrobutyricum* ATCC 25755 and selected homologs of other spore-forming Gram-positive strains. The phylogenetic tree was constructed by MEGA-X based on the distance method with Neighbor-Joining (NJ) with 1000 bootstrap replicates. The AbrB of *C. tyrobutyricum* is enclosed in red box.

**Figure 2 bioengineering-09-00575-f002:**
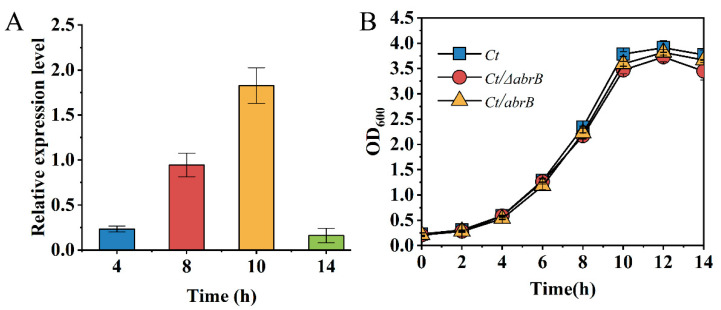
Characteristics analysis of AbrB in *C. tyrobutyricum*. (**A**) Growth phase-dependent *abrB* gene transcriptional profile; (**B**) Growth curves of *Ct*, *Ct/ΔabrB*, *Ct/abrB* and *Ct/ΔabrB-abrB* strains.

**Figure 3 bioengineering-09-00575-f003:**
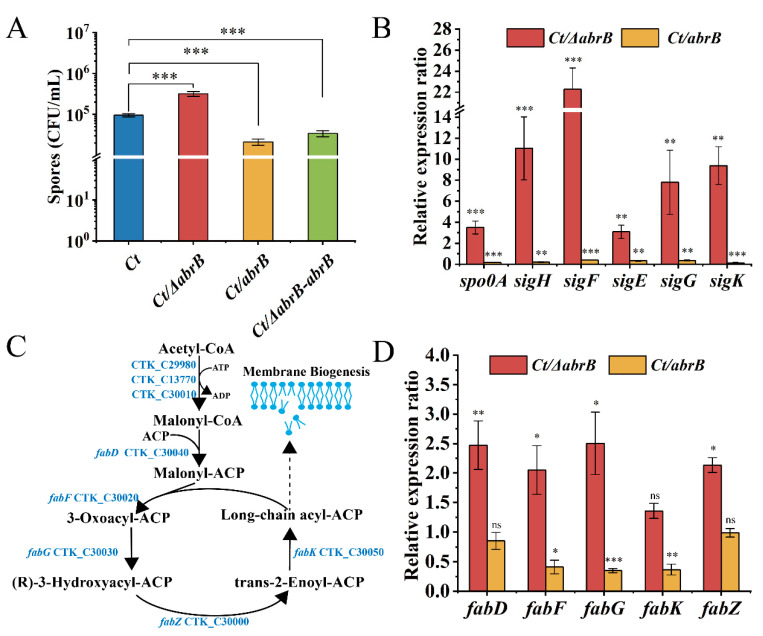
The effect of AbrB on sporulation in *C. tyrobutyricum*. (**A**) Sporulation of *Ct* and *abrB* mutants; (**B**) qRT-PCR analysis of sporulation formation related genes in *Ct/**ΔabrB* and *Ct/abrB* strains. Gene expression is represented as the fold difference normalized to *Ct*; (**C**) Lipid synthesis in *C. tyrobutyricum* based on KEGG database; (**D**) qRT-PCR of lipid synthesis-related genes in *Ct/**ΔabrB* and *Ct/abrB* strains. The acetyl-CoA C-acetyltransferase (*thl* gene) was used as the reference gene. The data are the means and standard deviations of at least two replicates. (ns, non-significant; *, *p* < 0.05; **, *p* < 0.01; ***, *p* < 0.001, *t*-test).

**Figure 4 bioengineering-09-00575-f004:**
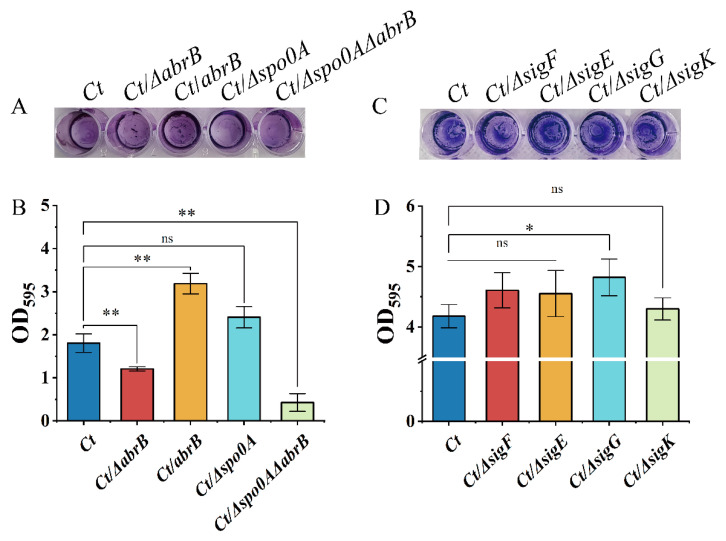
Multiple transcriptional regulators affect biofilm formation in *C. tyrobutyricum.* (**A**) Adhered biofilm formation of the *abrB* or *spo0A* mutants. The cells were cultured at 37 °C for 2 days and stained with crystal violet; (**B**) Quantification of the adhered biofilm biomass of *abrB* and *spo0A* mutants. (**C**) Adhered biofilm formation of the sigma mutants; (**D**) Quantification of the adhered biofilm biomass of the sigma mutants. Statistical significance was tested by one-way analysis of variance (ns, non-significant; *, *p* < 0.05 **, *p* < 0.01). The data are the means and standard deviations of three replicates.

**Figure 5 bioengineering-09-00575-f005:**
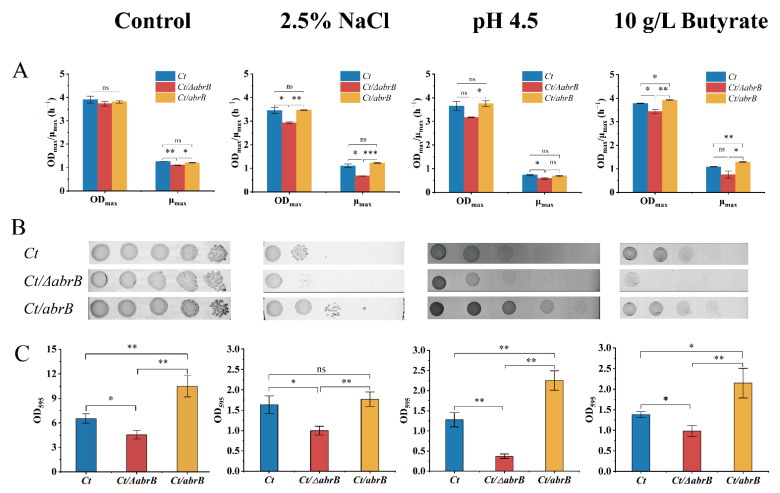
Deletion of *abrB* impairs the tolerance of *C. tyrobutyricum* to adverse environments. (**A**) The growth condition of *Ct*, *Ct/**ΔabrB* and *Ct/abrB* under different adverse environments; (**B**) Spot assays of *Ct*, *Ct/**ΔabrB* and *Ct/abrB*; (**C**) Adhered biofilm formation of *Ct*, *Ct/**ΔabrB* and *Ct/abrB.* (ns, non-significant; *, *p* < 0.05; **, *p* < 0.01, one-way ANOVA). The data are the means and standard deviations of three replicates.

**Figure 6 bioengineering-09-00575-f006:**
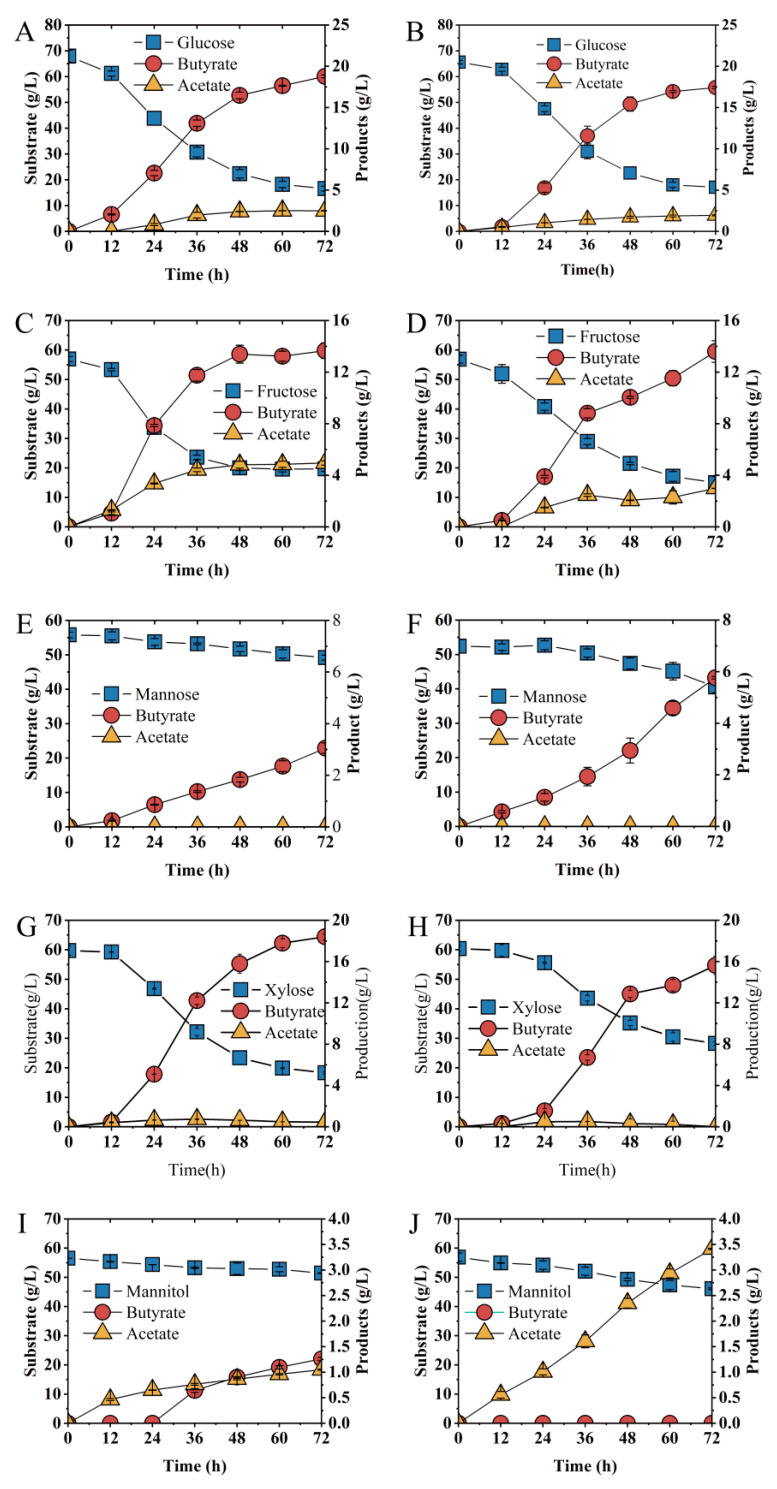
Fermentation kinetics of *Ct* (**left** row) and *Ct/**ΔabrB* (**right** row) under different substrates as sole carbon source in CGM supplement with CaCO_3_. Glucose (**A**,**B**), Fructose (**C**,**D**), Mannose (**E**,**F**), Xylose (**G**,**H**) and Mannitol (**I**,**J**) in CGM supplement with CaCO_3_.

**Figure 7 bioengineering-09-00575-f007:**
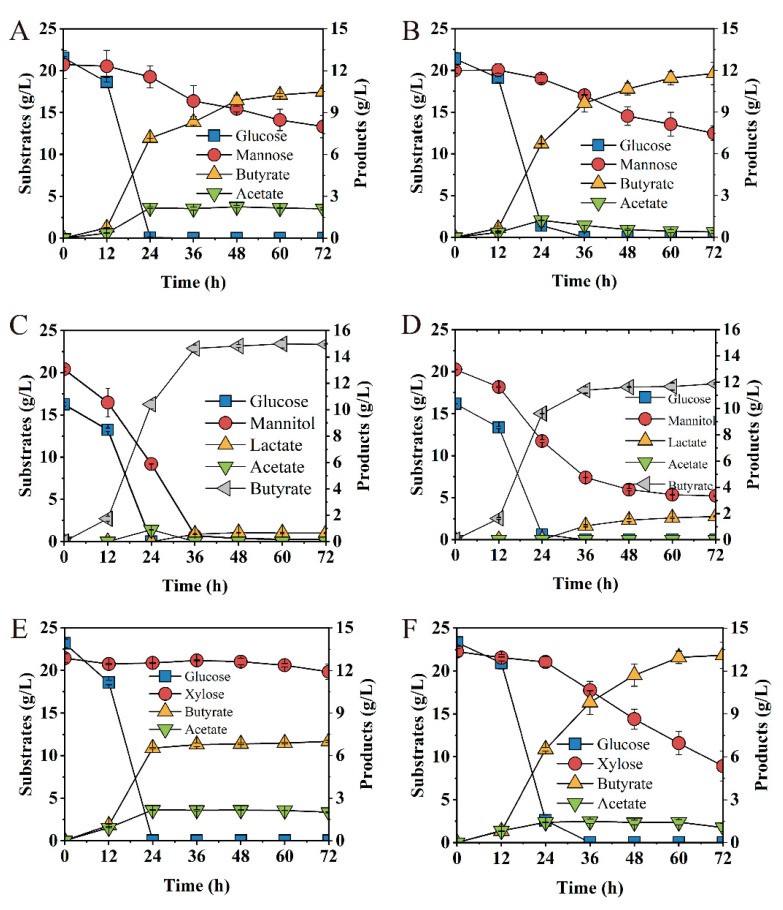
Fermentation kinetics of *Ct* (**left** row) and *Ct/**ΔabrB* (**right** row) with glucose/mannose mixture (1:1) (**A**,**B**), glucose/mannose mixture (1:1) (**C**,**D**), and glucose/mannitol mixture (1:1) (**E**,**F**) in CGM supplement with CaCO_3_.

**Figure 8 bioengineering-09-00575-f008:**
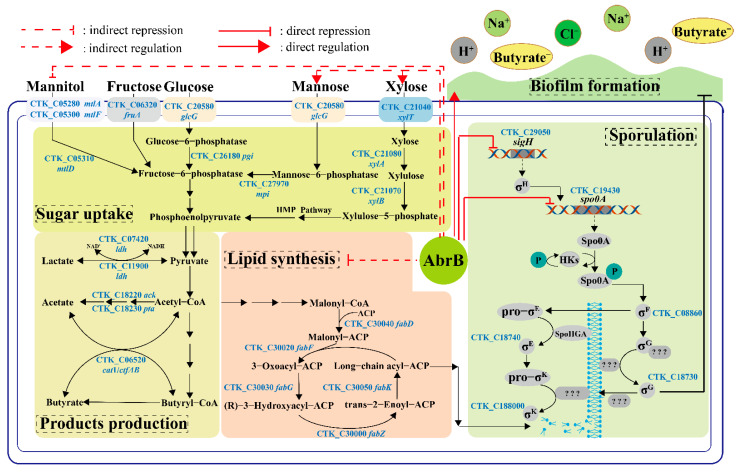
The possible regulatory mechanism of AbrB in *C. tyrobutyricum* ATCC 25755. Sugar uptake and products production: The utilization of xylose or mannose was repressed while the utilization of mannitol was enhanced when fermentation occurred with a sole carbon source. AbrB could regulate products distribution when mannitol was used as the carbon source. Sporulation and Lipid synthesis: AbrB repressed sporulation by binding to the promoter regions of *sigH* and *spo0A*. After *abrB* knockout, the expression of *sigH* and *spo0A* was up-regulated significantly. Meanwhile, *abrB* deletion alleviated the inhibition of the lipid synthesis pathway and provided raw materials for the synthesis of new cell membranes during asymmetric division and forespore engulfment; Biofilm formation: AbrB directly promoted biofilm formation and improved the tolerance of *C. tyrobutyricum* to adverse environments.

**Table 1 bioengineering-09-00575-t001:** Sugar consumption and products formation with different substrates in Ct and Ct/ΔabrB strains.

Substrate	Strain	Sugar Consumption Rate (g/L/h)	Butyrate		Acetate	
			g/L	g/L/h	g/L	g/L/h
Glucose	*Ct*	0.71 ± 0.05	18.78 ± 0.17	0.26 ± 0.00	2.49 ± 0.03	0.03 ± 0.00
	*Ct/ΔabrB*	0.67 ± 0.04	17.42 ± 0.14 *	0.24 ± 0.00 *	1.94 ± 0.06	0.03 ± 0.00
Xylose	*Ct*	0.58 ± 0.02	18.40 ± 0.28	0.26 ± 0.00	0.44 ± 0.00	0.00 ± 0.00
	*Ct/ΔabrB*	0.45 ± 0.01 **	13.71 ± 0.57 **	0.19 ± 0.01 **	-	-
Glucose/Xylose	*Ct*	0.97 ± 0.02(G)/ 0.02 ± 0.01(X)	7.01 ± 0.07	0.10 ± 0.00	2.02 ± 0.04	0.17 ± 0.00
	*Ct/ΔabrB*	0.65 ± 0.02(G)/0.19 ± 0.02(X) **	13.11 ± 0.92 *	0.18 ± 0.01 *	1.06 ± 0.28 *	0.09 ± 0.02 *

Statistical significance was tested with *t*-test (* *p* < 0.05 ** *p* < 0.01).

## Data Availability

All data generated or analyzed during this study are included in this published article and its Appendix A.

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
