# Peer review of "The Physiological Functions of AbrB on Sporulation, Biofilm Formation and Carbon Source Utilization in Clostridium tyrobutyricum"

_bioengineering, 2022, doi:10.3390/bioengineering9100575_

Round 1
Reviewer 1 Report
JournalBioengineering (ISSN 2306-5354) Manuscript ID bioengineering-1957762 Type Article Title The physiological functions of AbrB on sporulation, biofilm formation and carbon source utilization in Clostridium tyrobutyricumWhat was the purity and suppliers of chemicals added (chloramphenicol (Cm) and/or 50 μg/mL kan-amycin (Kan), thiamphenicol (Tm) and/or 40 μg/mL Erythromycin (Em)) ?
In part "2.9. Analytical methods",what is detector during the HPLC analysis ?
It is minor revision and paper could be published in Bioengineering after improvements made.
Author Response
Response to Reviewer 1 Comments
Point 1: What was the purity and suppliers of chemicals added (chloramphenicol (Cm) and/or 50 μg/mL kan-amycin (Kan), thiamphenicol (Tm) and/or 40 μg/mL Erythromycin (Em))?
Response: We have provided detailed information in Material and Methods part, shown in red font in page 3 lines 104-106.
Point 2: In part "2.9. Analytical methods", what is detector during the HPLC analysis?
Response: Sugars and organic acids were determined by HPLC (Waters, Milford, USA) with an Aminex HPX-87H Column (Bio-Rad, USA) and more information about the HPLC analysis has been provided in page 5 lines 194-196.

Reviewer 2 Report
Bearing in mind the fact that AbrB may play important role in a broad range of cellular processes as a global transcriptional regulator, Luo and collaborators focused on deciphering its pleiotropic regulatory function in Clostridium tyrobutyricum. Their manuscript entitled “The physiological functions of AbrB on sporulation, biofilm formation and carbon source utilization in Clostridium tyrobutyricum” is an effect of adequately designed studies which brought the authors to interesting conclusions. In general, the work was properly conducted and well presented. Therefore, I recommend publishing the manuscript provided that the authors address several minor issues listed below.
- In regard to figure 8 it is highly recommended that the authors distinguish direct (on sporulation) and indirect/undefined (on other processes) influence of AbrB, which could be done for example with solid vs dashed lines. Moreover, ArbB influence on “Sugar uptake” and “Products production” should also be depicted and the “CTKs” should be defined in the figure legend (if applicable).
- Some further suggestions for figures include: transformation of X axis to Fig.2A to a real timescale, application of discontinuous Y axes on Fig. 3A and 4D, one of Y axes titles on Fig. 5C is damaged.
- Also, some important methodological details are missing: how HPLC analysis was performed (system and column(s) used, solvents/mobile phase, temp. etc.), the meaning and way of calculation of ODmax and MUmax, gel electrophoresis and sample preparation (for sake of clarity a comment on results shown on Fig. S1D and S2B could also be added in figure legends)
- It would be also beneficial to clarify within the text what the “Ct/ΔabrB-abrB strains” (mentioned in figures 2 and S1) represent.
- Although the English quality is in general acceptable, a number of grammar (e.g. last sentence of the abstract should contain “regulatory”, p.2: “While deletion of abrB1941…”, in the last line on p. 8 should have “regulate”, p. 5 it is better to use “adverse environment” instead of “reverse environment” etc.) and typographical (e.g. p. 2 BLASTP should be capitalized, wrong bracket within ref. 36 on p.3 etc.) errors require correction. The authors should pay particular attention to scientific soundness (e.g. p.2 it is “surfactin” not “surfacing”, all the gene names – including e.g. “thl” on p.4 should be italic, reference to KEGG database is missing etc.).
Author Response
Response to Reviewer 2 Comments
Point 1: In regard to figure 8 it is highly recommended that the authors distinguish direct (on sporulation) and indirect/undefined (on other processes) influence of AbrB, which could be done for example with solid vs dashed lines. Moreover, ArbB influence on “Sugar uptake” and “Products production” should also be depicted and the “CTKs” should be defined in the figure legend (if applicable).
Response: Thank you for your advices. The content and legend of figure 8 have been modified according to your suggestions. The different fermentation results would be obtained with different carbon source, so it was difficult to depict in the figure 8 about the complex influence of AbrB on “Sugar uptake” and “Products production”. RNA-Seq or Electrophoretic mobility shift assay might be required to further determine the effect of AbrB on metabolism. (page 15 line 483)
Point 2: Some further suggestions for figures include: transformation of X axis to Fig.2A to a real timescale, application of discontinuous Y axes on Fig. 3A and 4D, one of Y axes titles on Fig. 5C is damaged.
Response: The timepoints for sample in Fig.2A is decided by cell growth. One timepoint means one group, the end of the lag phase(4h), middle (8 h) and the end (10 h) of the exponential phase, the declined phase(14h). According to your suggestion, the figures including Fig. 3A, 4D and 5C have been modified. (page 8 line 301, page 9 line 335 and page 10 line 368)
Point 3: Also, some important methodological details are missing: how HPLC analysis was performed (system and column(s) used, solvents/mobile phase, temp. etc.), the meaning and way of calculation of ODmax and μmax, gel electrophoresis and sample preparation (for sake of clarity a comment on results shown on Fig. S1D and S2B could also be added in figure legends)
Response: Thank you for your valuable comments. The missing part of the important methodology have been described and supplemented in detail. The comment in the results have been added to the figure legends. (page 3 lines 131-136, page 4 lines 159-161, page 5 lines 194-196)
Point 4: It would be also beneficial to clarify within the text what the “Ct/ΔabrB-abrB strains” (mentioned in figures 2 and S1) represent.
Response: Thanks for your advice and we have added the description of “Ct/ΔabrB-abrB strains” in the “Materials and Methods”. (page 3 lines 131-133)
Point 5: Although the English quality is in general acceptable, a number of grammar (e.g. last sentence of the abstract should contain “regulatory”(pleiotropic regulatory function of AbrB), p.2: “While deletion of abrB1941…”, in the last line on p. 8 should have “regulate”, p. 5 it is better to use “adverse environment” instead of “reverse environment” etc.) and typographical (e.g. p. 2 BLASTP should be capitalized, wrong bracket within ref. 36 on p.3 etc.) errors require correction. The authors should pay particular attention to scientific soundness (e.g. p.2 it is “surfactin” not “surfacing”, all the gene names – including e.g. “thl” on p.4 should be italic, reference to KEGG database is missing etc.).
Response: Thanks for your advice and we have carefully checked the manuscript and revised these grammatical, typographical errors with red font (page 1 line 23, page 2 line 58 and 89, page 3 line 108, page 4 line 180, page 9 line 325-326, page 10 lines 351-357, page 18 lines 644-645).
Reviewer 3 Report
Comments: The work is an interesting and smart manuscript, However there are few conceptual issues in this article, Would you please clarify following arguments?
1- Western blot should have been used to investigate phosphorylation of Spo0A
2- Authors should to provide full name for all abbreviations in text.
3- Refs should to be revised thoroughly according to format style of bioengineering MDPI journal.
4- Immunohistochemistry or immune assay should have been used to study the up/down regulation of σH and spo0A genes.
5- Uniform the scientific name of “Spo0A” or “spo0A” and “abrB” or “AbrB”.
6- Few English Typos Error should to be revised thoroughly over the text
Author Response
Response to Reviewer 3 Comments
Point 1: Western blot should have been used to investigate phosphorylation of Spo0A.
Response: Thank you for your valuable comments. We believe that understanding the phosphorylation of Spo0A is of great significance for studying the sporulation mechanism of C. tyrobutyricum, but our purpose is to explore the function of AbrB. Actually, the phosphorylation of Spo0A and the regulation of Spo0A~P on downstream sigma factors have been extensively studied in Bacillus and Clostridium. AbrB will not affect the phosphorylation of Spo0A.
Point 2: Authors should to provide full name for all abbreviations in text.
Response: Thank you for your useful advice. All abbreviations have been given full names for the first time in this article (page 1 lines 13, page 2 lines 64 and 67, page 5 line 221).
Point 3: Refs should to be revised thoroughly according to format style of bioengineering MDPI journal.
Response: Thanks for your advice and we have revised the “References” part carefully according to format style of the journal.
Point 4: Immunohistochemistry or immune assay should have been used to study the up/down regulation of σH and spo0A genes.
Response: Thank you for your valuable comments. Your comments provided good guidance for us to explore the sporulation mechanism of C. tyrobutyricum in further study.
Point 5: Uniform the scientific name of “Spo0A” or “spo0A” and “abrB” or “AbrB”.
Response: Thanks for your advice. “Spo0A” and “AbrB” represented proteins, while “spo0A” and “abrB” represented genes, which were scientific expressions.
Point 6: Few English Typos Error should to be revised thoroughly over the text.
Response: Thank you for your valuable comments. We have checked the manuscript carefully and revised these errors with red font.

Round 2
Reviewer 3 Report
Manuscript was revised point by point according to reviewer comments. Manuscript is more acceptable now